# Could Caregivers’ Stressful Care Experiences Be Related to Their Attitudes towards Advance Care Planning? A Cross-Sectional Descriptive Study

**DOI:** 10.3390/ijerph18179038

**Published:** 2021-08-27

**Authors:** Pei-Yu Tsai, Wen-Han Huang, Yu-Jun Chang

**Affiliations:** 1Department of Family Medicine, Department of Hospice-Palliative Care, Changhua Christian Hospital, Changhua 50006, Taiwan; 2Department of Hospice-Palliative Care, Changhua Christian Hospital, Changhua 50006, Taiwan; Hanksky0520@gmail.com; 3Big Data Center, Epidemiology and Biostatistics Center, Changhua Christian Hospital, Changhua 50006, Taiwan; 83686@cch.org.tw

**Keywords:** family caregiver, caregiver stress, care experience, advance care planning

## Abstract

Previous research has shown that care experiences influence the willingness for advance care planning (ACP). Family caregivers have increased contact with medical providers and procedures in the process of caring, and they have also witnessed the disability and suffering of patients. However, few studies have focused on family caregivers to understand their attitudes towards ACP. The aim of this cross-sectional study was to acknowledge family caregivers’ attitudes towards ACP and the related factors, especially care stress and experiences during the care process. We interviewed 291 family caregivers, and the demographics of the caregivers and care recipients, the clinical condition of care recipients, and the caregivers’ stress and care experiences were collected via anonymous questionnaires. Multiple logistic regression was performed to determine the factors associated with the attitudes towards ACP. We found that the caregiver having private health insurance (*p* < 0.001) and a completed DNR (*p* < 0.001) and the experience of recipients admitted to the ICU (*p* = 0.019) are associated with caregiver’s positive attitudes towards ACP. The greater the stress of conflict within a family over care decisions, the more participants think that ACP is important (*p* = 0.011). It is suggested that (1) in a family-centered culture, a public strategy for promoting ACP could be to emphasize the benefits of ACP in reducing family conflicts, and (2) when people make financial plans, they should also be provided with information about ACP to enable them to form a more integral plan for their future.

## 1. Introduction

Advance care planning (ACP) enables individuals to define goals and preferences for future medical treatment and care, to discuss these goals and preferences with family and healthcare providers, and to record and review these preferences if appropriate [1]. Even in Asian societies that emphasize family participation in an individual’s medical decision-making, ACP should be prospective and not be obstructed [2]. In Taiwan, the implementation of ACP is regulated by the “Patient Right to Autonomy Act” [3] which was legislated in late 2015. According to the “Patient Right to Autonomy Act”, the declarant, at least one relative of first- or second-degree affinity, and the health care agent shall engage in ACP. After ACP intervention, the declarant writes down the legal advance decision.

Prior studies have shown the factors that influence the individual acceptance of ACP. However, few studies have focused on family caregivers to understand their attitudes towards ACP. Due to caring for disabled relatives, family caregivers have more interactions with medical providers and are exposed to various medical interventions earlier than the general public, and they also witness the disability and suffering of care recipients. Do these experiences provoke these caregivers to think about their own values, and make them more willing to discuss their future medical treatment earlier?

Family members may experience various stressful situations during the care process. Stress may be associated with social–cultural variables (employment and education); interpersonal relationships including family responsibilities; and other factors, such as the severity of the recipient’s illness, duration of caring and the caregiver’s personal health status [4], and the financial burden [5,6]. It remains unclear if, and to what extent, caregivers’ stress is related to their attitudes towards ACP.

The purposes of this cross-sectional study were to acknowledge family caregivers’ attitudes towards ACP and the related factors, especially care stress and experiences during the care process.

## 2. Materials and Methods

### 2.1. Study Design and Setting

This study was designed as a cross-sectional descriptive study, and recruited family caregivers in central Taiwan, from November 2019 to February 2020. The Institutional Review Board of Changhua Christian Hospital (CCH) approved the study (CCH IRB No. 190909, approval date: 29 October 2019), and all participants signed informed consent. Interviewers were medical staff qualified in nationally recognized ACP consultation training courses or state-certificated core lecturers in the “Patient Right to Autonomy Act”. This ensured that each interviewer has a consistent and correct understanding of ACP. Before the formal interview, the research team members reached a consensus about the access process.

The sampling was conducted on general medical, geriatrics, and hospice wards; in home-based settings; in long-term care facilities; and at family-medicine or hospice-care outpatient clinics. To be eligible for inclusion in the study, participants had to have been a caregiver of their adult family members when the care recipients were admitted to wards or at home. The performance status of the care recipients was scored with the Eastern Cooperative Oncology Group (ECOG) performance status scale to assess the level of functioning in their daily living activities [7]. The score ranges from 0 to 5, indicating from full activity to death. Only the caregivers providing care for recipients with ECOG 2, 3, or 4 were included in this study, which is to say that the care recipient was restricted in physically strenuous activities. In cases where the care recipient was in a critical condition, their caregiver was excluded from participation.

Before the participant answered the questionnaire, the interviewer used the official ACP literature as an aid to explain ACP to the participant. Then, we assessed the participant’s knowledge of ACP with a three-true-false item test. If wrong answers were given, the interviewer immediately clarified the concepts for the participant. These three questions were reviewed by the experts of the ACP team and were found to be reflective of the subject’s understanding of ACP.

### 2.2. Measures

The ACP-attitude questionnaire, used to measure a participant’s attitude towards ACP, which was adapted from a pilot study [8], consists of four items: “ACP is important,” “I will engage in ACP,” “I support my family to engage in ACP,” and “I agree that ACP can relief family pressure.” A self-report 10-point Likert scale was used, where a higher score represents a more positive attitude of the subject. The full score is 40 points. The Cronbach’s alpha reliability coefficient for the valid samples collected in this research was 0.89 (>0.8).

### 2.3. Covariates

Demographic Information, Caregiver Stress, and Care Experiences

The participants’ personal demographic information solicited in this study included their age, gender, marriage status, education attainment, occupation, health status, private health insurance, relationship to care recipient, and completion of a “do not resuscitate” (DNR) document.

The care recipients’ personal demographic information solicited included their age, gender, three major diagnoses, completion of a DNR document, and ECOG performance status.

Care experiences, in this article, were defined as the matters that happened during the care process. These care experiences were divided into two groups: (1) (caregivers’) care load and (2) (care recipients’) suffering. The items in the care load group included the duration of caring, daily caring hours, and whether the caregiver was the medical decision-maker of a care recipient. The items in the suffering group included whether the care recipient had a tube (s) for any purpose, the severity of illness (terminally ill or not), and whether the care recipient had been admitted to the intensive care unit (ICU).

The caregiver’s stress was assessed using the Taiwanese version of the Kingston Caregiver Stress Scale (KCSS). The KCSS is primarily a scale to allow family caregivers to express the level of stress that they experience and has been widely used to assess the stress of caregivers of patients with various diseases [9,10,11]. The KCSS is a 10-item, 5-point self-rating scale where 1 means “no stress” and 5 means “extreme stress”. The KCSS includes three domains of stress: caregiving issues, family issues, and financial issues. The scale has good internal consistency and reliability both in the original version [12,13] and in the Taiwanese version (Cronbach’s alpha 0.89) [14].

Finally, referring to the method in Hu’s study [15], we used a semi-open questionnaire to ask participants why they are willing to engage in ACP (multiple options could be selected).

### 2.4. Statistical Analysis

In order to better understand the relevance of family caregivers’ care experiences to caregiving stress and their attitudes towards ACP, we conducted a series of bivariate analyses of caregivers’ demographic, psychosocial, and care recipient-related factors affecting caregiver stress and their attitudes towards ACP. The employed statistical tests included Student’s *t*-test or one-way ANOVA (for continuous variables) and Chi-square test or Fisher exact test (for categorical variables), as appropriate. In addition, we conducted the Spearman correlation analysis to measure the strength of association between caregiver stress and ACP-attitude. Finally, we employed the multivariable generalized linear regression model controlling possible confounding factors to explore those factors related to ACP-attitude. The final model only retained the significant predictors (*p* < 0.05). All data were analyzed using the IBM Corp. Released 2013. IBM SPSS Statistics for Windows, Version 22.0. Armonk, NY: IBM Corp. *p-*Value < 0.05 was considered statistically significant.

## 3. Results

### 3.1. Participant Demographics

In total, 300 caregivers were approached during the study period, although five caregivers indicated that they were unwilling to be interviewed, thus, the rejection rate was 1.67%. A total of 295 interviews were conducted. Four participants did not complete the questionnaire, meaning a total of 291 valid questionnaires were collected. One of the four participants refused to continue completing the questionnaire due to tiredness, and the three other interviews were interrupted as the participants had to deal with the patients’ physiological problems, resulting in a response rate of 98.6%. After the interviewers explained ACP, 93.5% of the participants on the first attempt correctly answered the three pre-designed questions about knowledge of ACP, which indicated that the participants had a correct understanding of ACP. The mean total score of ACP-attitude was generally high, with 33.4 out of 40 overall (SD = 7.9), among which, the four attitude items and the respective mean scores were as follows: “ACP is important,” 8.7 (SD = 1.8); “I will engage in ACP,” 7.7 (SD = 2.7); “I support my family to engage in ACP,” 8.3 (SD = 2.3); and “I agree that ACP can relieve family pressure,” 8.7 (SD = 2.1), and each score was out of 10. Overall, the participants of our study were aware of ACP and also showed positive attitudes.

The demographic background of the family caregivers and care recipients in this study are shown in Table 1. The mean ages were 49.9 and 76.9 years for the caregivers and care recipients, respectively. Most of the caregivers were children or children-in-law (63.9%), female (70.4%), post-high school educated (46.7%), healthy (69.1%), in the trade/service industry (47.1%), had private health insurance (84.9%), and had no completed DNR (88.3%). Most of the care recipients were female (53.3%), completely disabled i.e., ECOG performance score 4 (45.0%), and had a completed DNR (54.3%). The three major diagnoses were: cardiovascular diseases (44.7%, including heart failure, hypertension, and coronary artery diseases), cancer (43.3%), and organic brain disorder and cerebrovascular diseases (36.1%).

### 3.2. The ACP-Attitude Relevant to the Demographics of Caregivers Rather Than Recipients

The demographic backgrounds of the recipients were also related to the caregivers’ stress scores. (Table 1) The caregivers reported higher KCSS scores when the recipients were beyond 65 years old (*p* = 0.009), had a poor performance status (*p* = 0.010), or had a completed DNR (*p* = 0.003), but these factors did not affect the caregivers’ attitudes towards ACP. By comparison, those caregivers who were younger (*p* = 0.013), had received a high-level education (*p* = 0.035), had private health insurance (*p* < 0.001), or had a completed DNR (*p* < 0.001) had higher ACP-attitude scores. The caregivers being seriously ill had a negative correlation with their attitudes towards ACP (*p* = 0.004).

We further explored the reasons for the inclination of participants to engage in ACP (Figure 1) and found the top three reasons to be as follows: to avoid the burden on family members in decision making (70.4%), to express end-of-life care willingness (66.7%), and to avoid conflicts of opinions among family members (60.8%).

### 3.3. The Stress Mainly from Family Conflict Correlates with Caregivers Recognition of the Importance of ACP

To further determine the main stress factor relevant to the attitudes towards ACP, we explored the correlation between the various sources of the caregivers’ stress and their ACP-attitude (Table 2). The result shows that with a higher level of stress from conflicts among family members over care decisions, the participants were more likely to consider ACP to be important (Spearman’s *r* = 0.150, *p* = 0.011), and they demonstrated a slightly higher ACP-attitude score (*r* = 0.117, *p* = 0.046).

Additionally, the greater the stress about “concerns regarding the future care needs,” the more the participants thought that ACP is important (*r* = 0.120, *p* = 0.041), and they agreed slightly more that ACP can reduce the stress of family members (*r* = 0.116, *p* = 0.048). However, we found no significant correlation of the KCSS score with ACP-attitude (*p* = 0.173).

### 3.4. Various Care Experiences Cause Different Aspects of Stress, While Only the Care Recipients’ Former ICU-Admission Is Associated with the Caregivers’Positive ACP-Attitude

The relationship between caregivers’ care experiences and stress and their attitudes towards ACP are shown in Table 3. In the caregivers’ care load group, when the daily caring hours were more than 8 h, the KCSS scores were higher (*p* = 0.001), especially regarding stresses in the caregiving issue domain (*p* < 0.001) and in the financial issue domain (*p* = 0.011). Being a care recipient’s medical decision-maker also brought about higher scores in KCSS (*p* < 0.001), particularly attributed to both the caregiving (*p* < 0.001) and financial issues (*p* = 0.011). However, the duration of caring did not show a statistically significant difference in any of the caregivers’ stress domains.

In the recipient suffering group, when a care recipient was terminally ill, the caregivers reported higher scores in the caregiving issue domain, the family issues domain, and the total KCSS score (*p* < 0.001, *p* = 0.010, and *p* < 0.001, respectively). In instances where a recipient had been admitted to the ICU, the stress was higher in the caregiving issues domain (*p* = 0.001) and also in the total KCSS score (*p* = 0.003). In cases where a recipient had a tube(s), the stress was higher in the caregiving issues domain (*p* = 0.019) and in the total KCSS score (*p* = 0.021). In summary, every aspect of care experiences, except the duration of caring, puts caregivers under pressure from different domains. However, among the various stressful care experiences, only the experience of providing care for recipients who had been admitted to the ICU significantly increased the caregivers’ ACP-attitude score. (*p* = 0.019).

Controlling for possible confounding factors in the multivariable generalized linear model, a positive ACP-attitude was associated with caregivers with the following three conditions: seriously ill (95% CI: −14.589–−3.002; *p* = 0.003), with private health insurance (95% CI: 4.640–9.442; *p* < 0.001), and with a completed DNR (95% CI: 2.292–7.424; *p* < 0.001). ACP-attitude was also positively related to the care experience of caring for recipients who had been admitted to ICU (95% CI: 0.214–3.523; *p* = 0.027) (Table 4).

## 4. Discussion

This study aimed to understand family caregivers’ attitudes towards ACP and the related factors. The ACP-attitude is relevant to the demographics of caregivers rather than recipients. Caregivers’ stress mainly from family conflicts, correlates with their recognition of the importance of ACP. Various care experiences cause different aspects of stress, while only care recipients’ ICU -admissions are associated with caregivers’ positive ACP-attitude.

The results of our study indicated that the caregiver’s health status, private health insurance, and a completed formal DNR document are the main factors that affect their attitudes towards ACP. In our study, seriously ill caregivers showed more negative attitudes towards ACP than those who were healthy or had a chronic disease. The reason for this might be that the issues discussed in ACP about terminal illness, or fatal conditions, which may soon be confronted by seriously ill caregivers, could impose unpleasant feelings [16], leading to a negative attitude towards ACP. However, with the small sample size of the subgroup (*n* = 6) and the high variation in the ACP-attitude score, more samples are needed for further research.

Our study revealed that caregivers with private health insurance had a higher ACP-attitude score, which echoes the findings of a previous study in Australia, i.e., that undertaking a wider future planning process (e.g., making a will or financial enduring power of attorney) is a trigger for engaging ACP [17]. A reasonable explanation for this is that an insured person may have the trait or habit of planning ahead for important life occurrences, and is, therefore, more willing to plan for finance and health care issues in advance. Similarly, those who had a completed formal DNR had already put their future medical preferences into perspective and taken action to actualize them in the form of a document. Therefore, it is reasonable that DNR signees hold more positive attitudes towards ACP.

Similar to other studies [14,18], our research also found that the stress of family conflict is positively correlated with attitudes towards ACP; however, the correlation was weak. Analyzing the relationship between KCSS score and ACP-attitude, we observed that those caregivers who had experienced greater stress from conflicts within family members over medical decisions, assigned more importance to ACP. From the answers to the semi-open questionnaire, the main reasons for a caregivers’ willingness to engage in ACP were related to family concerns, such as: sparing loved ones the suffering of making difficult decisions and avoiding conflict of opinions. This finding adds another layer to the consensus of the current literature, proposing that the motivation for engaging in ACP is to avoid a burden on the family [19,20]. ACP is a process that encourages individuals and their families to express one another’s thoughts and emotions. Through communication and negotiation among patients and family members, a consensus can be reached, which can resolve possible family conflicts in the future [16,21]. Those caregivers who had experienced family conflicts during the care process were more likely to try to avoid exerting the same pressure on their family in the future, and would therefore consider the option of ACP and make care decisions for themselves in advance. Taiwan, similar to most other Asian countries, has a family-centered culture. Family-centered ethics posit that end-of-life care decisions are an intra-family matter, resulting in family members adopting uncertain attitudes towards ACP [21]. Most people are reluctant to admit conflicts in their families. A Chinese proverb states that a “harmonious family can lead to the success of everything.” Taiwanese people are worried that hidden intra-family conflict may come to the fore during the ACP process. Although they think ACP is important, they hesitate to engage in ACP (Table 3), and their attitudes towards ACP are relatively weakly positive. Understanding the cultural context of family conflict and acknowledging this issue during the ACP process can guide the declarant in improving communication quality.

Previous studies carried out in different cultural contexts have shown that witnessing the suffering of relatives and friends influences the willingness of caregivers to engage in ACP [8,18,19,20,22,23]. Chiang et al. [20] also pointed out that compared to the general public, family caregivers who provide care for disabled or seriously ill relatives are more positive and willing to accept the ACP concept. Our study further revealed that not all experiences are related to ACP-attitude. The results showed that care load experience has no correlation with ACP-attitude. In contrast, the experience of witnessing suffering is positively correlated with the attitudes towards ACP, specifically the suffering from ICU admission of care recipients, but not from recipients’ intubation (Table 3). In Chinese-based culture, reduced food intake or the inability to eat via the mouth are considered problematic. Most patients or family members would then choose artificial nutrition and hydration. According to data from the Ministry of Health and Welfare of Taiwan, up to 9.6% of hospitalized patients had a nasogastric tube inserted for enteral nutrition in 2020 [24]. A study reported that 29.2% of elderly people at long-term care facilities in Taipei, Taiwan, were fed through a tube [25]. This suggests that intubation for medical care is common and generally acceptable among Taiwanese people and explains why the presence of tubing is not related to the attitudes of caregivers towards ACP. In contrast, recipients’ suffering in the ICU has a greater positive impact on caregivers’ ACP-attitude. The exact reason for this needs further study.

Certain care experiences were associated with a higher KCSS score, especially in the caregiving issue domain. The study by Mok et al. found that Chinese people in Hong Kong, influenced by their culture and religion, accept the responsibility for taking care of an ill family member as a normal course of life or fate [26]. In Asia, many countries and regions are strongly influenced by Confucianism, Buddhism, and Taoism, such as China, Japan, Korea, Singapore, Hong Kong, and Taiwan. They share similar family ethics and filial piety. A common belief is that people who submit to fate will lead to a good life [27]. However, as found in this study, either from the care load or from witnessing the suffering of recipients, caregivers are under care pressure. Obviously, the statement that “caring for your family is your responsibility and fate” does not ease the pressure in the care process.

Financial factors might be a consideration in medical care decisions [28]. Financial stress is perceived once caregivers are also the medical decision-makers, especially when considering all of the consequences of medical care decisions, including the substantial burdens of logistic, finance, and care loads that may arise. For example, after treatment, patients could be disabled and bed-ridden for a long time, which often exacts a significant financial burden on their families.

When a patient is in the terminal stage of life, the caregiver’s family-issue stress increases. During this period, family members are often called upon to make end-of-life decisions, including whether to initiate, withhold, continue, or withdraw life-sustaining treatment. Families sometimes come into conflict, as multiple family members participate in the decision-making process [29], especially when they do not know the recipients’ wishes [30,31].

We acknowledge limitations to our study. First, we may need to take cultural context into account to properly interpret the results of this study. One of our findings is that compared to financial and caregiving issues, the ACP-attitude has a stronger association with the family-issue stress domain, which may not be analogous to other different cultures due to the intrinsic family-centered propensity of Taiwanese people. Second, we noticed that most of the participants showed a positive attitude towards ACP, possibly due to the “socially acceptable” response, rather than their true view of ACP [32]. This could have increased the ACP-attitude scores in general, thereby making it difficult for us to observe the differences in correlations between variates. Additionally, limited by the inherent disadvantage of a personal interview survey, one must be extremely cautious when interpreting attitudes towards ACP.

## 5. Conclusions

The study provided insights into family caregivers’ attitudes towards ACP and related factors. First, people with health insurance have more positive attitudes towards ACP. This result reminds us that when people undertake financial planning, they should also be provided with information about ACP to enable them to form a more integral plan for their future. Second, the results showed that care experiences, including the (caregivers’) care load and (recipients’) suffering, cause care pressure. However, no relationship was observed between the care load and ACP-attitude. On the contrary, ACP-attitude was positively related to the caregivers’ witnessing the patients’ suffering in the ICU. Therefore, for those whose family members have been admitted to ICU, we can provide them with information about ACP in a timely manner and encourage them to engage in ACP. When an ACP declarant mentions the experience of taking care of others as the motivation for engaging in ACP, it is necessary for the ACP team to further clarify the kind of stressful experience that has had that impact in order to conduct more efficient dialogues and to provide the full benefits of ACP. Third, caregivers facing stress from intra-family conflict acknowledge the importance of ACP more. This could be further supported by our finding that caregivers are significantly motivated to engage in ACP due to the expectations that ACP could help to relieve family suffering as a result of decision making and to avoid conflicts of opinions. In the clinic, we need to provide more support to medical decision-makers. Providing information about ACP to families with terminally ill relatives could be especially important in assisting them in making end-of-life medical decisions, as it was shown that these families experience the most stress from internal issues. In a family-centered culture, the benefits of ACP in reducing family conflict can be emphasized as a strategy for promoting ACP.

## Figures and Tables

**Figure 1 ijerph-18-09038-f001:**
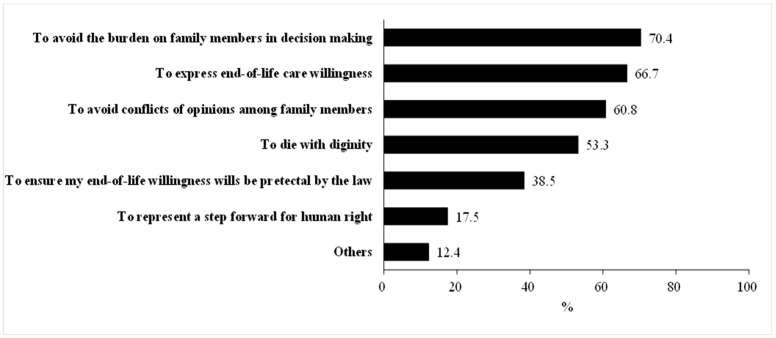
The reasons for willingness to engage in ACP (multiple options could be selected).

**Table 1 ijerph-18-09038-t001:** The relationship between the demographics of family caregivers/recipients and KCSS and ACP-attitude.

				Total KCSS Score			Attitude towards ACP		
		*n*	%	Mean	SD	*p*-Value	Mean	SD	*p*-Value
**Total**		291	100.0	20.2	7.3		33.4	7.9	
**Caregivers**									
Age	20–44	110	37.8	21.0	7.1	0.061	35.0	7.2	**0.013**
	45–64	139	47.8	20.2	7.4		32.8	8.0	
	≥65	42	14.4	17.9	7.5		31.1	8.9	
Gender	Male	86	29.6	19.8	7.0	0.555	32.5	7.9	0.210
	Female	205	70.4	20.3	7.5		33.8	7.9	
Health status	Healthy	201	69.1	20.6	7.4	0.364	34.2	7.5	**0.004**
	Has chronic disease(s)	84	28.9	19.3	7.2		32.0	8.4	
	Seriously ill	6	2.1	19.2	6.2		25.5	9.0	
Education	Elementary school	30	10.3	18.4	7.4	**0.032**	30.1	9.0	**0.035**
	Junior high school	42	14.4	21.2	7.3		32.1	8.1	
	Senior high school	83	28.5	21.8	7.8		33.5	8.4	
	Post-high school education	136	46.7	19.3	6.9		34.4	7.1	
Occupation	Farming fishery and Pasturage	17	5.8	18.5	4.9	0.818	30.4	9.5	0.443
	Public servant	23	7.9	20.2	7.0		34.6	6.9	
	Manufacturing	47	16.2	21.0	7.1		33.5	7.2	
	Trade/service industry	137	47.1	20.2	7.5		33.8	7.7	
	Housekeeper	67	23.0	19.9	8.0		32.8	8.7	
Private health insurance	Yes	247	84.9	20.0	7.1	0.271	34.5	7.0	**<0.001**
	No	44	15.1	21.3	8.5		27.3	9.8	
Relationship to patient	Spouse	43	14.8	21.8	8.6	0.094	33.1	7.9	0.484
	Child	129	44.3	20.6	6.9		33.3	7.9	
	Grandchild	34	11.7	16.6	5.4		35.1	8.0	
	Child in law	57	19.6	19.9	7.5		33.8	7.8	
	Sibling	6	2.1	20.2	8.3		28.0	9.5	
	Parent	18	6.2	20.8	7.7		31.6	8.2	
	Other	4	1.4	20.5	9.7		34.5	4.1	
Completion of DNR	Yes	34	11.7	20.1	7.6	0.924	38.3	3.2	**<0.001**
	No	257	88.3	20.2	7.3		32.7	8.1	
**Recipients**									
Age	20-44	6	2.1	22.7	10.2	**0.009**	30.8	8.2	0.324
	45-64	50	17.2	22.9	7.0		34.7	6.8	
	≥65	235	80.8	19.5	7.2		33.2	8.1	
Gender	Male	136	46.7	20.8	7.8	0.143	34.3	6.8	0.061
	Female	155	53.3	19.6	6.9		32.6	8.7	
Completion of DNR	Yes	158	54.3	21.4	7.5	**0.003**	33.8	7.6	0.353
	No	133	45.7	18.8	6.9		32.9	8.3	
ECOG	1	55	18.9	18.0	6.7	**0.010**	31.7	9.0	0.114
	2	57	19.6	19.4	7.1		34.2	6.3	
	3	48	16.5	22.7	7.6		35.2	6.5	
	4	131	45.0	20.5	7.4		33.1	8.4	

*p*-Value by Student’s *t*-test or one-way ANOVA when appropriate. *p*-Value < 0.05 was considered statistically significant.

**Table 2 ijerph-18-09038-t002:** The correlation between the caregiver stress and ACP-attitude.

	ACP Is Important		Willingness to Engage		Support My Family to Engage		Reduce Family’s Stress		Attitude towards ACP	
(*n* = 291)	*r*	*p*-Value	*r*	*p*-Value	*r*	*p*-Value	*r*	*p*-Value	*r*	*p*-Value
**Care giving issues**	0.059	0.314	0.008	0.885	0.050	0.393	0.048	0.414	0.058	0.322
Over-burdened	0.001	0.989	0.030	0.613	0.006	0.915	0.000	0.998	0.000	0.999
Change in your relationship	0.088	0.133	0.027	0.642	0.031	0.593	0.062	0.295	0.070	0.233
Changes in your social life	−0.028	0.630	−0.028	0.636	0.007	0.910	−0.024	0.682	−0.008	0.896
Conflicts with your previous daily commitments	−0.018	0.760	−0.022	0.706	0.006	0.921	0.020	0.734	0.015	0.793
Trapped by the responsibilities	0.064	0.276	0.013	0.827	0.048	0.413	0.040	0.493	0.061	0.298
A lack of confidence in your ability	0.040	0.493	−0.036	0.543	0.029	0.627	0.064	0.276	0.033	0.574
Concerns regarding the future care needs	0.120	**0.041**	0.025	0.677	0.104	0.077	0.116	**0.048**	0.108	0.065
**Family issues**	0.124	**0.035**	0.057	0.332	0.036	0.538	0.053	0.363	0.099	0.091
Conflicts within your family over care decisions	0.150	**0.011**	0.064	0.277	0.059	0.320	0.081	0.170	0.117	**0.046**
Conflicts over the amount of support	0.071	0.229	0.057	0.337	0.014	0.814	0.028	0.631	0.073	0.215
**Financial difficulties**	0.044	0.457	0.082	0.161	0.053	0.369	0.095	0.105	0.078	0.186
Financial issues	0.044	0.457	0.082	0.161	0.053	0.369	0.095	0.105	0.078	0.186
**Total KCSS score**	0.077	0.188	0.019	0.751	0.056	0.345	0.058	0.323	0.080	0.173

*r*: Spearman’s correlation coefficient.

**Table 3 ijerph-18-09038-t003:** The relationship between the caregivers’ care experience and stress and their attitude towards ACP.

			Care Giving Issues *			Family Issues **			Financial Issues ***			Total KCSS Score			Attitude towards ACP		
Care Experience		*n*	Mean	SD	*p*-Value	Mean	SD	*p*-Value	Mean	SD	*p*-Value	Mean	SD	*p*-Value	Mean	SD	*p*-Value
**Total**		291	14.6	5.6		3.8	1.9		1.7	1.1		20.2	7.3		33.4	7.9	
**Caregivers’ care load**																	
Duration of caring	<1 year	103	15.1	6.1	0.211	3.7	1.9	0.260	1.8	1.1	0.057	20.6	7.9	0.139	34.2	7.7	0.117
	1–3 years	50	15.4	5.7		4.2	2.1		2.0	1.1		21.6	7.8		34.6	7.5	
	>3 years	138	14.0	5.2		3.7	1.9		1.6	0.9		19.3	6.7		32.4	8.2	
Daily caring hours	< 8 h	149	13.2	5.0	**<0.001**	3.8	1.9	0.249	1.6	0.9	**0.011**	18.6	6.7	**0.001**	33.3	7.9	0.824
	8–16 h	52	16.2	6.2		4.1	1.9		1.9	1.2		22.2	8.0		34.0	7.0	
	>16 h	90	16.0	5.7		3.6	1.9		2.0	1.1		21.5	7.5		33.1	8.5	
Patient’s medical decision maker	Yes	161	15.9	5.5	**<0.001**	4.0	2.1	0.073	1.9	1.1	**0.001**	21.8	7.2	**<0.001**	33.1	8.0	0.568
	No	130	13.1	5.4		3.6	1.8		1.5	0.9		18.2	7.1		33.7	7.8	
**Recipients’ suffering**																	
Terminally ill	Yes	146	15.9	5.7	**<0.001**	4.1	2.1	**0.010**	1.8	1.1	0.103	21.8	7.5	**<0.001**	34.0	7.1	0.192
	No	145	13.4	5.3		3.5	1.7		1.6	1.0		18.5	6.8		32.8	8.7	
Has been admitted to ICU	Yes	133	15.8	5.7	**0.001**	4.0	2.1	0.058	1.8	1.2	0.727	21.6	7.4	**0.003**	34.6	7.6	**0.019**
	No	158	13.7	5.4		3.6	1.8		1.7	0.9		19.0	7.1		32.4	8.1	
Has tube(s) for any purpose	0	135	13.6	5.5	**0.019**	3.6	1.7	0.272	1.7	1.0	0.524	18.9	7.1	**0.021**	32.9	7.9	0.353
	1	86	14.9	5.6		3.8	2.0		1.7	1.0		20.5	7.3		33.2	7.8	
	2	57	16.1	5.3		4.2	2.3		1.8	1.1		22.1	7.2		35.1	7.4	
	≥3	13	16.4	7.3		4.2	1.7		2.1	1.6		22.7	8.9		32.4	10.5	

*p*-Value by one-way ANOVA or Student’s *t*-test when appropriate. *p*-Value < 0.05 was considered statistically significant. * Caregiving issues have scores ranging from 7 to 35. ** Family issues have scores ranging from 2 to 10. *** Financial issues have scores ranging from 1 to 5.

**Table 4 ijerph-18-09038-t004:** The results of generalized linear models on the attitude towards ACP.

		Bivariable Analysis (Crude)						Multivariable Analysis (Adjusted)					
		Estimate	SE	95% CI			*p*-Value	Estimate	SE	95% CI			*p*-Value
**Caregivers**													
Age		−0.105	0.032	−0.168	-	−0.042	0.001						
Health status	Seriously ill	−8.714	3.217	−15.020	-	−2.408	0.007	−8.795	2.956	−14.589	-	−3.002	**0.003**
	Chronic disease	−2.238	1.009	−4.215	-	−0.260	0.027	−0.514	0.973	−2.420	-	1.392	0.597
	Healthy	0.000						0.000					
Number of chronic diseases		−1.356	0.616	−2.564	-	−0.149	0.028						
Education	Post-high school	4.300	1.573	1.218	-	7.383	0.006						
	Senior high school	3.373	1.661	0.117	-	6.628	0.042						
	Junior high school	1.962	1.864	−1.691	-	5.615	0.292						
	Elementary school	0.000											
Private health insurance	Yes	7.178	1.225	4.778	-	9.579	<0.001	7.041	1.225	4.640	-	9.442	**<0.001**
	No	0.000						0.000					
Completion of DNR	Yes	5.588	1.407	2.831	-	8.345	<0.001	4.858	1.309	2.292	-	7.424	**<0.001**
	No	0.000						0.000					
**Recipients**													
Has been admitted to ICU	Yes	2.179	0.923	0.371	-	3.987	0.018	1.868	0.844	0.214	-	3.523	**0.027**
	No	0.000						0.000					

## Data Availability

The datasets generated and analyzed during the current study are not publicly available due to maintenance of confidentiality and privacy requirements consistent with the Institutional Review Board approval.

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
