# Peer review of "Could Caregivers’ Stressful Care Experiences Be Related to Their Attitudes towards Advance Care Planning? A Cross-Sectional Descriptive Study"

_ijerph, 2021, doi:10.3390/ijerph18179038_

Round 1
Reviewer 1 Report
Several sections were difficult to follow and need editing to provide clarity, specifically. Lines: 44-57,66-68, 297-301 and 331-341. The limitation section starts with a run-on sentence.
Stated purpose of the study in the abstract and lines 69-72 are different.
Methods:
Inclusion of participants: was time the caregiver had provided care considered? Would certainly influence response to questionnaires.
Explain ECOG acronym.
Explaining ACP to participants prior to question administration seems to introduce significant bias. Measures of importance of ACP are not reliable, as the participant would be hesitant to tell "the explainer" that ACP is not important. Perhaps explains the high importance and understanding rate.
Discussion section
Alignment with the referenced Chinese study is not clear. Are care giving beliefs similar among Chinese and Taiwanese?
Data for the statement about financial stress is not provided. (lines 259-263)
Line 271, what is the population of the Amjad et al. study? This information should be reported given, cultural influence is a key element of the research.
Reviewer 2 Report
Dear authors,
Congratulations on your work here. Considering caregiver's stress and caring circumstances improves and impacts the quality of life of patients.
I would like to recommend your work to be published, but with a few previous and minimal recommendations to change:
Abstract: is correct, but including two or three lines of further introduction before objectives could be better for readers' understanding.
Line 45: please check socio-culture, maybe socio-cultural variables could be a better expression.
Line 143: please provide this cite: IBM Corp. Released 2013. IBM SPSS Statistics for Windows, Version 22.0. Armonk, NY: IBM Corp.
Figure 1: please check the figure, line 4, the line needs to be centered.
Results, discussion, limitations, and conclusions are well supported and explained.
Author Response
Dear Reviewer,
Good Day!
In order to make the manuscript better reflect the research content and easier to read, the title of our manuscript has been changed to ‘Could Caregivers’ Stressful Care Experiences be Related to Their Attitudes towards Advance Care Planning? A Cross-sectional Descriptive Study’, and the order of paragraphs in this manuscript was also adjusted. We also revised our manuscript according to your comments. The revised manuscript has undergone English language editing by MDPI, and is now being submitted again to International Journal of Environment Research and Public Health to be considered for publication.
Please see the attachment for detailed responses.

Reviewer 3 Report
Thankyou for the opportunity to review this manuscript, which describes a cross-sectional descriptive survey of Taiwanese caregivers, regarding their caregiving experiences, caregiving stress and the relationship of these factors to their attitudes towards advance care planning. The authors make an important contribution in terms of investigating the nuanced ways that caregiving experiences can influence advance care planning attitudes. With that said I do have some concerns about the current presentation and have listed some comments and suggestions below.
General Comments:
1. My primary concern is with the statistical approach, which is built around a large number of separate bivariate tests and correlations between individual scale items, and a final multivariable regression model to adjust the analysis for potential confounding variables. One of the things that appeared to be implicit in the authors' study design was that caregiving experiences are acting on attitudes to advance care planning, in part by an indirect effect on stress (as measured by the KCSS). Given this it could be argued that a latent variable, structural equation modelling approach would be more appropriate. However this may not be possible with the data collected or the resources available. At the least I think the authors should consider the possibility of an inflated Type I error rate from multiple tests, either through corrected p values or through clearly acknowledging this in the limitations. The exploration of indirect effects would, I believe, be an interesting avenue for future research or secondary analysis of these data.
2. Another assumption implicit in the analysis is that it is the current caregiver experiences that are influencing the person's attitudes towards advance care planning. However previous research has focused more on previous experiences of providing care at the end of life. So I wonder if in fact many of these caregivers have already previously provided end of life care to other family members, and their attitudes have formed through this. Is there a way of stratifying the sample to separate those who have previously provided end of life care for a loved one versus those who haven't? A person's attitudes can be formed by their whole life story, not just their current experience.
Specific Comments:
1. Table 1 - should the p value for the effect of caregiver education on KCSS score be in bold face like the other p values?
2. L232 - remove the gendered term 'himself'
3. L240 (and throughout) - the analysis conducted is in fact a multiple or multivariable generalised linear regression model, not a multivariate model. Multivariate is the term used when there are multiple dependent variables being tested in the model.
4. L302 - should 'ethnics' be 'ethics'?
5. L309 - I'm not sure of the meaning of the term 'adverting'
6. L350 - 'stuffs' - this word is unclear here and does not fit
Author Response

(The authors gave the same response as above.)

Round 2
Reviewer 3 Report
The authors have satisfactorily addressed the comments I made previously